# Regulating Death and Disease: Exploring the Roles of Metacaspases in Plants and Fungi

**DOI:** 10.3390/ijms24010312

**Published:** 2022-12-24

**Authors:** Nalleli Garcia, Rachel E. Kalicharan, Lisa Kinch, Jessie Fernandez

**Affiliations:** 1Department of Microbiology and Cell Science, Institute of Food and Agricultural Sciences, University of Florida, Gainesville, FL 32611, USA; 2Howard Hughes Medical Institute, University of Texas Southwestern Medical Center, Dallas, TX 75390, USA

**Keywords:** metacaspases, programmed cell death, plant immunity, plant pathogens, pathogenicity, fungal pathogens

## Abstract

Identified over twenty years ago and distantly related to animal caspases are a group of cysteine proteases known as metacaspases. Throughout the years, much like caspase roles in metazoans, metacaspases have been shown to be involved in regulating cellular death in non-metazoan organisms. Yet, continued research on metacaspases describes these proteins as intricate and multifunctional, displaying striking diversity on distinct biological functions. In this review, we intend to describe the recent advances in our understanding of the divergence of metacaspase functionality in plants and fungi. We will dissect the duality of metacaspase activity in the context of plant-pathogen interactions, providing a unique lens from which to characterize metacaspases in the development, immunity, and stress responses of plants, and the development and virulence of fungi. Furthermore, we explore the evolutionary trajectory of fungal metacaspases to delineate their structure and function. Bridging the gap between metacaspase roles in immunity and pathogenicity of plant-pathogen interactions can enable more effective and targeted phytopathogen control efforts to increase production of globally important food crops. Therefore, the exploitation and manipulation of metacaspases in plants or fungi represent new potential avenues for developing mitigation strategies against plant pathogens.

## 1. Introduction

Plant pathogens are responsible for devastating agricultural losses across a wide array of crops, exacerbating the burden of global food insecurity [1]. Successful plant pathogens must bypass plant defenses while plants must employ a vast range of controlled defense mechanisms to subvert potential pathogenicity [2,3]. This complex interaction entails utilizing a variety of mechanisms, such as pathogen effector proteins and host resistance genes [4]. Intriguingly, a common denominator observed in these plant-pathogen interactions is the implementation and regulation of cellular death. Plant immune responses rely on precise and regulated cell death to impede pathogen invasion while ensuring the protection of the surrounding host cells [5]. Conversely, cell death is an important aspect in the maintenance of pathogen fitness, as it enables the pathogen to effectively cause disease and circumvent defense strategies executed by the plant host [6].

Initially, the regulation of cell death was only known in metazoan organisms. It was found that programmed cell death (PCD), the precise removal of unwanted cells following specific signals, in animals was coordinated by caspases [7,8]. Caspases are proteins found within the family of cysteine proteases. Their name is derived from their composition and amino acid specificity: cysteine aspartic-specific proteases [9]. Given the highly regulated network involved in PCD, caspases are initially synthesized as zymogens, or inactive enzymes. Caspases consist of a prodomain located at the N-terminus, followed by the p20 and p10 subunits. Upon certain stimulation, caspases undergo proteolytic auto-activation to form the active subunits of p10 and p20, which can then cleave substrates primarily in an aspartic acid-dependent manner [10]. While the identification of caspases progressed our understanding of cell death in metazoan organisms, it raised questions about the cell death mechanisms employed by other organisms. At the turn of the century, a group of proteins that had caspase-like characteristics were identified and named metacaspases [11]. These metacaspases were found only in protists, fungi, and plants. While these proteins could not be classified as caspases as their sequences were too diverged, metacaspases retained some structural homology to caspases that could suggest these proteins shared a common ancestor at one point. Metacaspases have since been discovered in deep-branching bacteria and archaea, supporting the notion that PCD machinery traces back to the emergence of cellular life [12,13,14,15]. There currently are three distinct types of metacaspases that have been reported in different eukaryotic organisms (Figure 1). In contrast to animal caspases, metacaspases primarily cleave their substrates in an arginine or lysine specific manner [16,17]. Most recently, the functional divergence of non-metazoan metacaspases in comparison to animal caspases has been well reviewed, highlighting the critical differences both proteases possess in a cellular and molecular context [18].

Over the years, metacaspases have emerged as intricate proteins, identified in modulating both plant immune responses and enhancing fungal cellular viability. As such, metacaspases provide a unique avenue to explore plant pathogen interactions as these versatile proteins shed light into the divergence of functionality between plant and fungal metacaspases and their crucial roles required for immune responses or pathogenicity. This review aims to synthesize the most recent findings of the multifunctionality of metacaspases in the scope of plants and plant pathogens to better characterize the dual role they play during plant infection and identify how both plant and fungal organisms are benefited or hindered by metacaspases. Specifically, we review the metacaspase functionality in plant immunity, development, and stress responses as well as fungal pathogenicity, development, and evolution. Future research conducted to better characterize metacaspases may identify effective mitigation strategies against plant pathogens potentially through priming the plant’s immune response, or by dysregulating important pathogen cellular functions.

## 2. Plant Metacaspase Expression following Pathogen Invasion

As sessile organisms, plants must rely on cell signaling to perceive threats, alert neighboring cells of incoming attacks, and activate immune defenses [21,22]. In the context of pathogen infections, plants utilize various modes of defense to interfere with the spread of invading pathogens. The initial, basal line of plant defense comes from plant receptors recognizing generally conserved structural components of plant pathogens, known as pathogen-associated molecular patterns (PAMPs), thus triggering PAMP-triggered immunity (PTI) [2,23]. If phytopathogens can bypass this initial defense via secretion of effector proteins that suppresses PTI, plants can deploy a more precise and effective immune response in an effort to prevent pathogens from colonizing additional cells. This second line of defense is known as effector-triggered immunity (ETI) [23]. ETI is able to activate the plant hypersensitive response (HR), which is a form of cell death that is rapidly triggered to contain the pathogen and thereby prevent further colonization [5]. HR has been observed for many years and is characterized by a combination of vacuolar cell death and necrosis features [24,25,26,27]. During pathogen attacks, the plant HR is mediated through a gene-for-gene interaction with specific pathogen proteins known as avirulence (Avr) genes that can be recognized through plant resistant (R) genes [28]. When R genes recognize certain Avr genes within the cell, immune responses are initiated to prevent pathogen proliferation and propagation of infection. Many plant R genes code for nucleotide-rich leucine-binding receptor (NLR) proteins that help transmit an immunity signal cascade following binding to pathogen effectors [29,30]. The deployment of HR requires the activity of specific plant proteases to coordinate effective and regulated cell death. Such proteases identified to regulate HR are threonine proteases and cysteine proteases like metacaspases [31].

As such, the expression of various plant metacaspases has been linked to pathogen invasion. In tomato, the type II *LeMCA1* metacaspase was found to be upregulated following infection with the fungal pathogen, *Botrytis cinerea* [32]. Additionally, *LeMCA1* is positioned near genes involved in PCD, suggesting a potential role in pathogen-triggered PCD [32]. Likewise, overexpression of the type II *Nicotiana benthamiana* metacaspase *NbMCA1* was seen following infection with the fungal pathogen *Colletotrichum destructivum* [33]. Interestingly, silencing *NbMCA1* showed reduced resistance to *C. destructivum*, but no changes in PCD were observed, suggesting a potential non-death role for NbMCA1 in modulating plant immunity [33]. Yet, in pepper plants, the silencing of the type II metacaspase *Camc9*, a potential metacaspase that was shown to be overexpressed during bacterial infection of *Xanthomonas campestris pv. vesicatoria*, caused a decrease in disease resistance and suppression of cell death symptoms [34]. Similarly, a wheat type II metacaspase, TaMCA4, was shown to play a positive role in promoting PCD during *Puccinia striiformis f. sp. tritici* fungal infection [35]. An overexpression of *TaMCA4* in wheat showed increased PCD levels, while the silencing of *TaMCA4* showed reduced necrotic lesions after infection with an avirulent *Pst* strain. Interestingly, the avirulent *Pst* strain elevated *TaMCA4* expression levels, but no change in *TaMCA4* expression was seen when using a compatible *Pst* strain [35]. Additionally, it was only with the silencing of *TaMCA4* that the avirulent *Pst* strain was able to infect wheat, suggesting that TaMCA4 could be a major player in preventing certain *Pst* strains from becoming virulent in wheat [35]. Years later, a wheat type I metacaspase, *TaMCA1*, was found to be upregulated when wheat was infected with a compatible *Pst* strain [36]. When *TaMCA1* was silenced, less cell death was observed and an accumulation of H_2_O_2_ was seen [36].

Through numerous plant species, there is a pattern of increased metacaspase gene expression following pathogen infections, suggesting a vital role for metacaspases in promoting immune defenses albeit through different processes. The regulation of plant immunity may come in different forms, either through regulating plant PCD or, like NbMCA1, through other mechanisms independent of PCD. Additionally, it is important to note that calcium is required for most plant metacaspase activity [37,38]. Interestingly, recent data shows that during physical damage of *Arabidopsis thaliana*, glutamate is also important for stress signaling, binding to glutamate receptor-like (GLR) proteins in order to alter the concentrations of Ca^2+^ in the cell, which quickly travel through the plasmodesmata or the vasculature from wounded cells to promote defense responses [39] (Figure 2). This is one of the first reports characterizing long distance stress signaling by glutamate through manipulating Ca^2+^ concentrations in the cell. It will be intriguing to see if this change in Ca^2+^ flux due to the release of glutamate contributes to metacaspase activation during mechanical wounding, and if this mechanism is conserved through other plant species to trigger PCD within the cell following mechanical damage.

This damage-mediated Ca^2+^ flux may function as a way in which the plant can quickly sense external cues and promote internal signaling cascades, thereby regulating PCD. We are only beginning to understand the interplay between plant immunity and metacaspases following plant pathogen exposure, so investigating metacaspases in this context could help answer critical questions. For instance, could pathogen virulence factors suppress plant immune responses by specifically interfering with glutamate activity? Are metacaspases and their potential activation pathways specific targets for pathogens? Our further understanding of plant metacaspase involvement in plant immunity comes truly from identifying the cellular interactors of metacaspases.

### 2.1. What Are Plant Metacaspases Cleaving?

While an overexpression of metacaspases and regulation of PCD can be seen in various plant organisms after pathogen infection, less is known about the substrates targeted by plant metacaspases. During PTI, the plant is also able to sense internal host molecules known as damage-associated molecular patterns (DAMPs) to trigger defense responses [40]. Plant elicitor peptides (Peps) are DAMPs that can be released during pathogen invasion [41]. Peps themselves are produced via the cleavage of their precursor proteins, PROPEPs, which is a component of the plant innate immune response [42]. Previously, it was unknown what exactly regulated the production of Peps during PTI. An in vitro study on the metacaspase AtMC4 from *A. thaliana* showed AtMC4 to be important for Pep1 maturation from PROPEP1, and this interaction promoted defense responses in the plant following mechanical damage to plant tissue [43] (Figure 2). A subsequent study done with *A. thaliana* protoplasts confirmed the cleavage of PROPEP1 by AtMC4 [44]. This in-depth study also reported that the bacterial elicitor flg22 increased PROPEP1 expression and promoted AtMC4-mediated cleavage of PROPEP1, most likely due to flg22-induced Ca^2+^ accumulation which further promoted AtMC4 activity [44]. Together, these studies demonstrate the interconnection between plant immunity mechanisms and metacaspase regulation. AtMC4 is able to provide a direct pathway for plant defense responses following both mechanical damage and pathogen invasion through controlling the rate of Pep1 production. Furthermore, the upregulation of *AtMC4* in response to mechanical damage poses the question if the level of metacaspase expression bears a direct relationship to the degree of damage sustained by the plant host. More specifically, such an investigation can reveal if there is cross talk between the PAMP and DAMP pathways, or if certain proteins are strictly confined to specific pathways/means of compromise.

Most recently, an *A. thaliana* metacaspase was found to be involved in helping trigger PTI [45]. The plasma membrane-anchored receptor-like cytoplasmic kinase PBS1-LIKE 19 (PBL19) normally is tethered to the plasma membrane. Following chitin recognition by the receptor LYSM-CONTAINING RECEPTOR-LIKE KINASE 5 (LYK5) and co-receptor CHITIN ELICITOR RECEPTOR KINASE 1 (CERK1), PBL19 is translocated to the nucleus via its nuclear localization signal (NLS). Through the cleavage of the NLS of PBL19 by AtMC4, the metacaspase-cleaved PBL19 alters its localization from the nucleus to the cytoplasm [45]. This change in localization allows for the metacaspase-cleaved PBL19 to phosphorylate a known regulator of PCD, ENHANCED DISEASE SUSCEPTIBILITY 1 (EDS1), and in turn, amplify plant defense signals [45] (Figure 2). This finding expands on the functionality of plant metacaspases that are involved in enhancing plant immunity against plant pathogens through altering the localization of immune system regulators. EDS1 has also previously been linked to promoting ETI in *A. thaliana* [46]. Further characterizing the overlap of EDS1 in immunity may help bridge the gap between the two major immune responses in plants.

Additionally, recent research has also shown links between metacaspase activity and ETI. In grapevine, two metacaspases, *VrMC2* and *VrMC5* were found to be upregulated following infection with the pathogen *Plasmopara viticola* [47]. Interestingly, in grapevine lines overexpressing either *VrMC2* or *VrMC5* showed HR induction with the bacterial elicitor, harpin [47]. These cell death events were shown to be mediated by ROS bursts in the apoplast facilitated by a plasma membrane-located NADPH oxidase [47]. These results demonstrate potential downstream pathways that could be involved in metacaspase activation to promote immune related responses.

Furthermore, the *A. thaliana* metacaspases, AtMC1 and AtMC2, had previously demonstrated a role in regulating PCD following *Pseudomonas syringae* infection through either positively or negatively controlling pathogen-triggered HR via NLR interaction, respectively [48]. It is known that activation of NLR is directly associated with the upregulation of defense-related genes [30]. Therefore, metacaspase association with NLR mechanisms seems to be crucial for apt PCD in response to pathogens. This was validated when research conducted on maize (*Zea mays*) metacaspases provided further insight into the way in which metacaspase activity mediates HR through the modulation of a maize R gene coding for an NLR protein, Rp1-D21 [49]. It was shown that two type I metacaspases, ZmMC1 and ZmMC2, were involved in suppressing Rp1-D21-mediated HR in *N. benthamiana,* likely through an interaction with the coiled-coil domain of Rp1-D21 [49] (Figure 2). When Rp1-D21 was infiltrated via *Agrobacterium*-mediated transient expression without ZmMC1 or ZmMC2, *N. benthamiana* leaves showed HR symptoms, but the symptoms were reduced following the co-expression of ZmMC1 or ZmMC2 and Rp1-D21 [49]. Additional studies showed ZmMC1 and ZmMC2 altered Rp1-D21 subcellular localization from a uniform distribution in the nucleocytoplasm to a more punctate co-localization with ZmMC1 and ZmMC2 [49]. It was found through co-immunoprecipitation and yeast-two-hybrid assays that the physical association of Rp1-D21 N-terminal coiled-coil (CC_D21_) with ZmMC1 and ZmMC2 promoted the regulation of HR. In all, ZmMC1 and ZmMC2 have been shown to act similar to AtMC2 in *A. thaliana* given their functions in negatively regulating plant immune responses [48,49]. While maize seems to contain 11 different metacaspases, only ZmMC1 and ZmMC2 were shown to have this negative regulation to HR through their interaction with Rp1-D21. Other metacaspases that did not interact with Rp1-D21, such as ZmMC9, did not colocalize with Rp1-D21 or negatively affect cell death symptoms. While there was no evidence of cleavage or lytic activity from ZmMC1 or ZmMC2, the researchers suggest ZmMC1 or ZmMC2 work instead to aggregate Rp1-D21 inside of the cell, preventing it from localizing in a different compartment that may trigger HR.

Surprisingly, the autophagy marker, Atg8a, colocalizes in the punctate localizations along with ZmMC1 and ZmMC2 suggesting that both metacaspases are localized in autophagosomes [49]. The researchers thus speculate that Rp1-D21 is translocated to autophagosomes through interactions with ZmMC1 and ZmMC2, and Rp1-D21 is potentially taken to the vacuole for degradation to inhibit HR [49]. This finding is interesting given that previous investigations showed AtMC1-mediated PCD to act independently of autophagy pathways, suggesting that there are parallel pathways involved in mediating pathogen-triggered HR in plants [50]. Further studies need to be conducted in order to better understand if the suppression of HR is done through ZmMC1 and ZmMC2 regulating the localization of Rp1-D21 and acting in association with the autophagy pathway. Moreover, the *N. benthamiana* infiltrations of ZmMC1, ZmMC2, and Rp1-D21 were conducted without any calcium supplementation. Either Ca^2+^ levels within the plant were enough to promote activation of ZmMC1 and ZmMC2, or aggregation of Rp1-D21 did not require metacaspase activation and would point to additional functionalities of metacaspases that are calcium-independent. The presence of multiple pathways to tightly regulate metacaspase activity and PCD demonstrates their critical nature to host viability and homeostasis, as well as host adaptability and versatility to various situations that would require metacaspase-mediated PCD.

### 2.2. Plant Development, Stress Responses, and Cell Death

The roles of plant metacaspases expand beyond plant immune responses. While not in the scope of this review, we will lightly highlight metacaspase functionality in plant development and stress responses. For instance, numerous metacaspases identified in grapes showed increased expression during ovule development [51]. Two metacaspases from *Populus* trees have also reportedly been involved in cell death of xylem elements [52]. A study done on the metacaspase mcII-Pa from Norway spruce (*Picea abies*) demonstrated that during plant embryogenesis, mcII-Pa translocated from the cytoplasm to the nucleus to promote vacuolar cell death through DNA fragmentation [53,54]. A follow up study conducted on mcII-Pa demonstrated that in the absence of mcII-Pa or autophagy activity, previously expected vacuolar cell death activity shifted to necrotic cell death [55]. It seems that in *P. abies*, mcII-Pa is essential to maintain PCD through vacuolar cell death activity and, in its absence, PCD mechanisms switch to necrosis. Further questions arise on the specificity that metacaspases may have to certain forms of plant PCD and how their absence or overexpression may impact plant viability, particularly during cellular developmental stages, under certain stress conditions, or during pathogen attacks.

The *A. thaliana* metacaspase, AtMC8, was reported to exhibit sensitivity to H_2_O_2_ and UVC stress; exposure of these stressors when overexpressing *AtMC8* resulted in increased levels of PCD [56]. AtMC4 was also shown to be associated with increased sensitivity to a PCD inducer and oxidative stress inducers [57]. The characterization of metacaspases in rice, tomato, and rubber tree showed changes in metacaspase expression levels during increased salt or drought conditions [58,59,60]. This could indicate that metacaspases are one of the primary modulators of controlling PCD as a means to regulate damage done from external stimuli. A recent transcriptome analysis identified *Sedum alfredii* Metacaspase-1-like (AMC1) protein to be sensitive to heavy metal compounds [61]. Following indium treatment on wheat seedlings, indium-toxicity promoted wheat root cell death through the upregulation of metacaspases *TaMCA1* and *TaMCA4* [62]. Through a variety of environmental cues, plant metacaspases appear to be involved in a variety of stress responses that aim to protect the plant from undesirable external conditions. How a variety of metacaspases can be triggered through several different stressors remains to be understood. Further dissecting the interplay between metacaspase activated PCD and autophagy may bring additional networks to light that mediate metacaspase activity from stress responses.

## 3. The Basis for Fungal Metacaspases

The first metacaspase was found in the genome of *Saccharomyces cerevisiae,* which encodes a singular metacaspase known as Yca1 [11,63]. Like caspases, Yca1 has also been found to be involved in regulating cell death and oxidized proteins in the cell during stress conditions [64]. These findings illuminated a key player involved in the regulation of cell death in fungi. Especially of importance is the N-terminal prodomain of Yca1, which was shown to be needed for the clearance of insoluble aggregates [65]. Recent work has also shown an association between Yca1 and the ubiquitin-proteasome system (UPS) [66]. It was determined that the interaction between an E3 ligase (Rsp5) and Yca1 impacted the role of Yca1 in regulating both cellular vacuolar response and the insoluble aggregate levels in the cell [66]. Ubiquitination of Yca1 at K355 and phosphorylation at S346 promoted the interaction with Rsp5 [66]. Moreover, it was reported that Yca1 cleaved a ubiquitin precursor protein, Rps31, to promote the release of ubiquitin in the cell, in essence functioning as a deubiquitinating enzyme, aiding in maintaining proper ubiquitin concentrations that positively influences the clearance of protein aggregation [66]. This study is the first to establish how Yca1 can influence free ubiquitin in the cell to promote its association with Rsp5 in order to maintain the proteosome in the cell through regulating autophagy and protein aggregation. The interconnection between cellular processes and metacaspases demonstrates the adaptable nature of these proteins and gives insight into how some metacaspases can be responsible for their own activation. Studies in filamentous fungi are necessary to confirm if these interactions extend beyond single cell fungi, which could point to additional associations between metacaspase regulation of the UPS and autophagy pathways to promote adequate proteasome levels in the cell.

### 3.1. Fungal Metacaspase Involvement in Pathogenicity

Though the identification of metacaspases in plant pathogens are scarce, recent studies characterizing novel metacaspases in *Usilago maydis* and *Magnaporthe oryzae* depict metacaspases as multifaceted proteins involved in pathogen fitness and promoting disease in fungal plant pathogens following Ca^2+^ activation [67,68]. *U. maydis* is a biotrophic fungus responsible for smut disease in maize. The only metacaspase in *U. maydis*, Mca1, has been found to have a role in aiding vegetative growth, executing PCD, promoting the clearance of insoluble aggregates during stress conditions, as well as maintaining proper pathogenicity in maize [67]. The two metacaspases found in the hemibiotrophic fungus *M. oryzae*, MoMca1 and MoMca2, depict similar roles for maintaining proper fitness of the fungus to promote disease [68]. It was found that calcium activated MoMca1 and MoMca2 promoted the clearance of insoluble aggregates and were needed for full pathogenicity in rice [68]. Future studies are needed to explore the ways in which these metacaspases maintain the *M. oryzae* proteosome. This could point to a potential association between *M. oryzae* metacaspases and the UPS as was shown with Yca1, which would further explain how *M. oryzae* is able to promote the clearance of insoluble aggregates. While this study did not show MoMca1 and MoMca2 to be involved in PCD, it did suggest a potential role in stress response as the double metacaspase knockout strain showed increased radial growth on plates following stress tests [68]. The increased radial growth in the absence of MoMca1 and MoMca2 could suggest these metacaspases act as negative regulators of cell death, or perhaps their functionality is independent of PCD. These observations, in conjunction with the diminished stress response, indicate these proteins wield functional duality in vegetative growth and pathogenicity. Furthermore, the localization of MoMca1 and MoMca2 may give potential clues as to additional functions of these metacaspases to see if they are involved in important fungal differentiation stages needed for pathogenicity as was suggested in the delayed development of *M. oryzae* in the absence of both metacaspases. This delayed development could signify issues with autophagic cell death which is needed for proper growth and infection [69]. Deeper examination may uncover additional relationships between filamentous fungi metacaspases and autophagy in promoting pathogen development and, in effect, pathogenicity. As plant metacaspases work towards strengthening plant immunity and stress responses, plant pathogen metacaspases promote cell fitness to cause disease. The branching of these roles in different organisms demonstrates the complexity of metacaspase proteins especially as they relate to immunity and disease.

### 3.2. Cell Death Roles of Fungal Metacaspases

Cell death in fungus, much like that of plants, differs from normal animal cell death due to its cell wall and lack of caspase proteins. Characteristics of fungal cell death include DNA fragmentation, cellular shrinkage, increased ROS production, and upregulation of metacaspase activity [70]. Metacaspase activity has been documented in fungi as it relates to ageing and stress responses. For instance, in *Podospora anserina,* two metacaspases, *PaMCA1* and *PaMCA2*, were shown to be upregulated in older cultures, suggesting their involvement during senescence [71]. Additionally, *PaMCA1* expression was increased when cultures were treated with H_2_O_2_ [71]. These observations substantiate the multifunctional capacity of fungal metacaspases, delineated in their ability to operate in development, as well as in stimuli and stress responses. Moreover, the role of metacaspases in fungi can be paralleled to that of plant metacaspases, verifying that conserved functions of the protein are not necessarily restricted to a single kingdom.

In non-pathogenic fungi, poly ADP-ribose polymerases (PARPs) have been identified to be targeted by two metacaspases in *P. anserina* [72]. A PARP-like protein was also shown to be targeted and degraded during sporulation of *Aspergillus nidulans* [73]. PARPs have been identified as mediating DNA repair in the cell that has been damaged due to stresses [74]. This is similar to mechanisms seen with animal cell caspases, cleaving PARPs and promoting cell death [75]. A similar substrate was identified for both *S. cerevisiae* and the protozoan parasite *Trypanosoma brucei* [76]. Both *S. cerevisiae* and *T. brucei* were shown to cleave in vitro the DNA-damage inducible protein 1 (Ddi1), another evolutionary conserved eukaryotic shuttle protein that interacts with the UPS. In yeast, the cleavage of Ddi1 by Yca1 only occurred after increasing Ca^2+^ concentrations in the cell [76]. Expanding on this interaction may yield more information on the ways in which metacaspases interact with the UPS to promote proteolytic events to maintain the proteosome in the cell. Additional characterization of metacaspase substrates in other fungal species will further illustrate the distinct cell death mechanisms employed by fungi during stress conditions.

### 3.3. Structure and Evolution of Fungal Metacaspases

The first metacaspase crystal structure, solved for a metacaspase of the protist *T. brucei* (TbMCA2), provided insight to several key functional aspects of the family and its divergence from caspases [77]. TbMCA2 does not require dimerization or proteolytic processing into p20 and p10 subunits for activity. The core caspase fold of TbMCA2, which includes elements of the p20 and p10 subunits, was extended by a β-hairpin insertion from the variable linker that stabilized the enzyme as a monomer (Figure 3A). Invariant aspartic acid residues mark both the Ca^2+^ binding site required for activation, as well as the predicted P1 binding pocket determining specificity for arginine. The N-terminal proline-rich sequence of the prodomain spans the active site, potentially acting as a gatekeeper that regulates substrate entry. Removal of the TbMCA2 prodomain by proteolytic processing occurs in vitro and seems to be required for optimal activity and substrate specificity [78,79]. Thus, the metacaspase fold appears to be built for multiple modes of regulation, which provide a rich source for acquiring evolutionary diversity.

Several additional metacaspase structures have expanded our understanding of their regulation. The yeast type I metacaspase Yca1 structure is like that of TbMCA2, with an extra β-hairpin blocking the dimerization surface of the caspase sheet. Yca1 exhibits similar specificity for peptide substrates with arginine or lysine, and activation by Ca^2+^. However, Yca1 appears to be autoproteolytically processed in the linker region following positively charged residues [19]. The Yca1 structure lacks much of the low complexity prodomain (begins at residue 90), and the remaining sequence does not block entry to the active site as seen for TbMCA2. Thus, the prodomain appears to possess alternate functions in these metacaspases. The more recent structure of the type II metacaspase AtMC4 from *A. thaliana* was solved in an inhibitory conformation [38]. A region from the long linker that characterizes the type II metacaspases blocks Ca^2+^ activation and substrate access to the active site. The linker includes a helical domain insertion in its C-terminus that is responsible for its large size. The AtMC4 linker sequence covers the dimerization site with a loop (as opposed to a β-hairpin in the other two structures), and Ca^2+^-activation involves multiple sites of cleavage in the linker, leading to conformation change that promotes substrate processing (Figure 3B).

The type I metacaspases found in fungi (Figure 1) exhibit diversity in sequence and structure [11]. Type I metacaspases are also present in primitive organisms such as eubacteria, archaea, and phytoplankton, which lack the N-terminal prodomain and possess a short linker region [11,12,15]. Sequence analysis of the archaeal metacaspases highlight their preservation of the catalytic HC dyad, the S1 pocket, and Ca^2+^ binding site (Figure 3C). However, two of the conserved S1 binding pocket residues exhibit diversity in the archaeal sequences [12]. The potential roles of fungal metacaspases in pathogenicity suggest their divergence could result from a host-pathogen arms race, where the metacaspases can adopt multiple alternate forms of activation.

The experimental structures of the type I and II metacaspases provide a basis for understanding the observed sequence divergence of the family and its evolutionary relationship to caspases. The TbMCA2 and Yca1 β-hairpin insertions extend the core sheet of the caspase fold, resulting in a monomeric metacaspase enzyme. This β-hairpin insertion belongs to the variable linker observed among metacaspase sequences, and its position forming the edge of the β-sheet at the surface allows replacement with different structure elements. This diversity is observed in the type II metacaspase structure, which uses a loop to cover the caspase dimerization surface. The mobile nature of the linker suggested by a lack of electron density in the Yca1 and AtMC4 structures might also contribute to the ability of the metacaspase family to drive diversity. Such mobility allows recognition of different linker sequences for auto-processing, permits replacement of the prodomain with the linker as an autoinhibition feature, and provides flexibility for the conformation change associated with Ca^2+^ activation.

Overall, the conserved functionality of these metacaspases, considering the sequence variations and deviations observed, further substantiates the vital role they wield in fungal fitness. Moreover, monitoring and analyzing how metacaspases change over time can help identify potential metacaspase properties that are distinct to a certain species, as well as better characterize metacaspase structure and function on a molecular basis. Investigation of the downstream consequences of metacaspase sequence evolution and deviation may even uncover novel interactions with other pathways or elements ultimately associated with fungal development, stress response, PCD regulation, and/or pathogenicity.

## 4. Future Perspectives

It is becoming increasingly evident the vital role metacaspases hold in plant-pathogen interactions. On the plant side, metacaspases can be critical in the prevention of pathogenicity, while on the fungal side, they can promote evasion of host defenses. Plant metacaspases have been more extensively researched, giving insight into their wide range of functions in development, immunity activation, stress response, and PCD regulation. Follow-up studies on these discoveries could reveal interactions of metacaspases with other pathways or compounds involved in regulating these functions. Similarly, while Yca1 has led to many identifications of metacaspase homologs in higher fungal species, it will be interesting to see if future studies find additional associations between plant pathogen metacaspases and developmental stages, ageing, or interactions with the UPS. Examining the evolutionary trajectory of fungal metacaspases can further illuminate additional functionalities or even functional redundancies.

In a relationship that is constantly described as an arms race, it will be fascinating to continue dissecting the roles of metacaspases within plant-pathogen interactions. For instance, do pathogen metacaspases counteract plant metacaspase activity? Does enhancing metacaspase activity in plants or inhibiting fungal metacaspases prevent disease proliferation? Through an agricultural lens, including and considering the influence of metacaspases on both ends of the disease can lead to more robust and concentrated mitigation strategies. Approaches that focus on the influence of metacaspases in disease incidence and prevention can provide a molecular basis of understanding virulence and defense mechanisms, ultimately increasing the production rate of globally relevant food sources. Metacaspases are undoubtedly a critical factor in plant-pathogen interactions, and going forward, diving deeper into the incidence and evasion of plant disease must also mean exploring the roles and involvement of metacaspases.

## Figures and Tables

**Figure 1 ijms-24-00312-f001:**
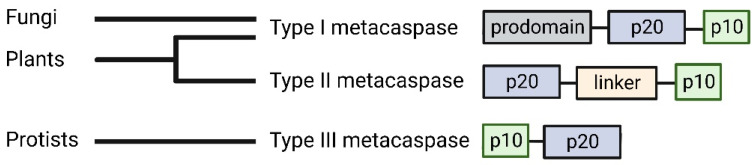
Type I metacaspases share structural similarities to caspases in that they contain an N-terminal prodomain (grey) which also includes proline-rich repeats and zinc finger motifs. Type II metacaspases lack a prodomain and contain a linker (light orange) over five times longer than caspases between the p20 (blue) and p10 (green) subunits [15,19]. Type III metacaspases have been found in phytoplankton and are characterized through the inverted sequential organization of the active subunits, with the p10 subunit situated closer to the N-terminus than the p20 subunit [15,20]. Created on BioRender.com on 6 December 2022.

**Figure 2 ijms-24-00312-f002:**
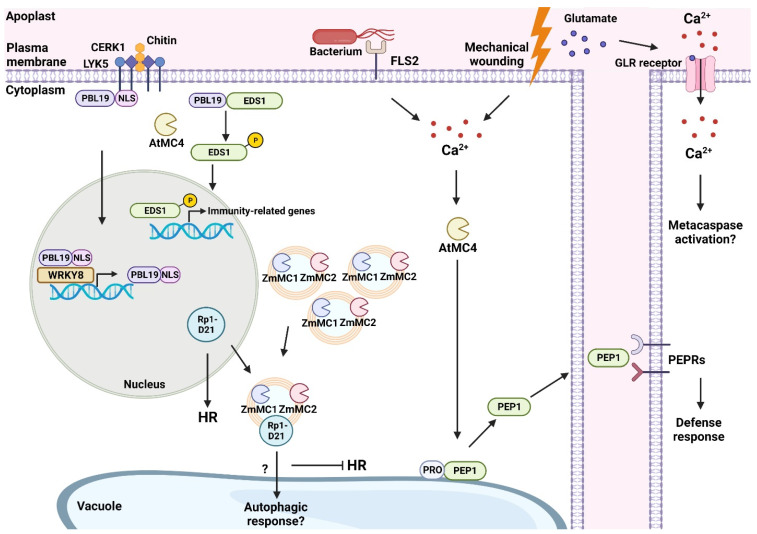
This working model depicts some plant metacaspases from *A. thaliana* and maize that are involved in promoting immune responses following physical damage or pathogen invasion. In *A. thaliana*, CERK1 (CHITIN ELICITOR RECEPTOR KINASE 1) and LYK5 (LYSM-CONTAINING RECEPTOR-LIKE KINASE 5) receptors triggered by chitin promote PBL19 (PBS1-LIKE 19) translocation to the nucleus. In the nucleus, PBL19 can self-amplify using the WRKY8 transcription factor. The metacaspase AtMC4 cleaves the NLS (nuclear localization signal) of PBL19, causing an accumulation of PBL19 in the cytoplasm. Cytoplasm-localized PBL19 can phosphorylate EDS1 (ENHANCED DISEASE SUSCEPTIBILITY 1), which promotes the transcription of immunity-related genes once phosphorylated EDS1 enters the nucleus. Mechanical wounding or bacterial flagellum activating FLS2 (flagellin sensing 2 receptor) causes an influx of Ca^2+^ inside the cytoplasm. This accumulation of Ca^2+^ activates AtMC4 which can process PROPEP1 into PEP1 and alter its localization from the tonoplast to the cytoplasm. PEP1 can then bind to PEPRs located at the plasma membrane of healthy cells to activate defense responses in neighboring cells. Mechanical wounding in *A. thaliana* has been shown to promote glutamate-based long-distance signaling to neighboring, undamaged cells. Glutamate binds to GLR (glutamate-like receptor) proteins that open and facilitate an influx of Ca^2+^ and may promote further metacaspase activation. In maize, metacaspases ZmMC1/2 follow a punctate localization in autophagosomes. The NLR protein, Rp1-D21, resides in the nucleus and can promote HR in maize. The interaction between ZmMC1/2 and Rp1-D21 causes Rp1-D21 to translocate from the nucleus to autophagosomes. The potential degradation of Rp1-D21 causes a sequestering of HR. Created on BioRender.com on 6 December 2022.

**Figure 3 ijms-24-00312-f003:**
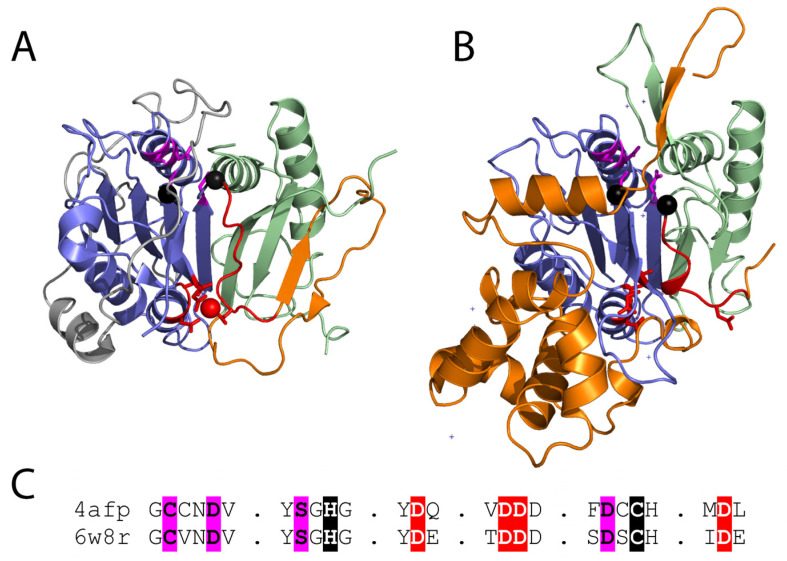
(**A**) Metacaspase from *T. brucei* (PDB 4afp) bound to a calcium mimetic (red sphere) with coordinating residues (red sticks) highlights the Ca^2+^ activation loop (red cartoon) that is followed by a variable sequence (orange cartoon) of different lengths that adopts diverse conformations in fungal metacaspases. Active site residue positions (black spheres) are near the P1 binding site (magenta sticks). The p20 and p10 subunits are colored blue and green, respectively, and the N-terminal extension is colored white. (**B**) Inactive AtMC4 structure (PDB 6w8r) is colored as in panel A. Multiple inserts to the structure are in white. (**C**) Corresponding motifs from the experimental structures are aligned. Functional residues are highlighted: Ca^2+^ binding (red), catalytic (black), and P1 pocket (magenta).

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
