# Peer review of "Regulating Death and Disease: Exploring the Roles of Metacaspases in Plants and Fungi"

_ijms, 2022, doi:10.3390/ijms24010312_

Round 1

Reviewer 1 Report

Very well written manuscript. Information on metacaspases is correctly presented. Synthetically collected information may contribute to the development of research on the importance of metacaspases. This manuscript will facilitate further analysis of the roles of metacaspases in plant-pathogen interactions.

Author Response

Response to Reviewer 1 Comments

Very well written manuscript. Information on metacaspases is correctly presented. Synthetically collected information may contribute to the development of research on the importance of metacaspases. This manuscript will facilitate further analysis of the roles of metacaspases in plant-pathogen interactions.

We thank Reviewer 1 for their efforts in reviewing our manuscript. We very much appreciate the feedback on our work.

Reviewer 2 Report

This review is an excellent work focusing on the metacaspase activity and plant-pathogen interaction. This study will have a practical implication in controlling disease incidence in plants. However, the author should revise the whole manuscript more carefully and scientifically, ensure the flow of information provided, and try to connect the current knowledge and research gap more stringently. Some parts need to be rewritten.

Paragraph three of the introduction seems not connected with the second paragraph—the flow of information is missing. The introduction section needs more background.

Line 68-70: Rewrite the sentence clearly.

Line 77-81: These sentences are like you are writing a research article. Please, carefully rewrite the sentences.

Write all the gene names in italics form in the manuscript.

Line 145-152: Break down and rewrite the sentence to make it clear. Make the research question, current knowledge, and future investigation prospects clear.

Author Response

Reviewer 2 Comments

This review is an excellent work focusing on the metacaspase activity and plant-pathogen interaction. This study will have a practical implication in controlling disease incidence in plants. However, the author should revise the whole manuscript more carefully and scientifically, ensure the flow of information provided, and try to connect the current knowledge and research gap more stringently. Some parts need to be rewritten.

Paragraph three of the introduction seems not connected with the second paragraph—the flow of information is missing. The introduction section needs more background.

We thank Reviewer 2 for their efforts in reviewing our manuscript. The critical feedback is much appreciated to make this manuscript as clear as possible. We have carefully considered the manuscript as a whole and made several changes based on these suggestions. Please find additional information added at the end of the second introductory paragraph (lines 59-62) to provide more structure in the manuscript.

Line 68-70: Rewrite the sentence clearly.

We thank Reviewer 2 for their editorial feedback. We have revised the sentence in question in lines 71-73 from “First observed in Saccharomyces cerevisiae with cell death functionalities, metacaspases have since emerged as intricate proteins, identified in modulating both plant immune responses and enhancing fungal cellular viability.” to “Over the years, metacaspases have emerged as intricate proteins, identified in modulating both plant immune responses and enhancing fungal cellular viability.”

Line 77-81: These sentences are like you are writing a research article. Please, carefully rewrite the sentences.

We thank Reviewer 2 for their feedback on the implications for these sentences. We agree and have applied changes to the sentences in lines 80-85. Please find the revised section changed from “Specifically, we will examine the metacaspase functionality of plant immunity, development, and stress responses as well as fungal pathogenicity, development, and evolution. Through better characterization of metacaspases, we may be able to identify effective mitigation strategies against plant pathogens potentially through priming the plant’s immune response, or by dysregulating important pathogen cellular functions.” to “Specifically, we review the metacaspase functionality in plant immunity, development, and stress responses as well as fungal pathogenicity, development, and evolution. Future research conducted to better characterize metacaspases may identify effective mitigation strategies against plant pathogens potentially through priming the plant’s immune response, or by dysregulating important pathogen cellular functions.”

Write all the gene names in italics form in the manuscript.

We thank Reviewer 2 for their feedback in our inconsistencies in formatting gene names. We have corrected this formatting issue in the manuscript with gene names in italics.

Line 145-152: Break down and rewrite the sentence to make it clear. Make the research question, current knowledge, and future investigation prospects clear.

We thank Reviewer 2 for their input in recognizing the ways in which this section could be clarified. Please refer to lines 157-163 for an updated version. We have since changed the sentences from “This could explain one of the ways in which external cues can be quickly sensed by the plant and used to promote internal signaling cascades to regulate PCD. While we are only recently beginning to understand the mechanisms of plant immunity that are being modulated by these metacaspases when exposed to plant pathogens, future investigations that characterize these metacaspases and their potential activation pathways could help determine if they are specific targets for pathogens. For instance, could pathogen virulence factors suppress plant immune responses by specifically interfering with glutamate activity?” to “This damage-mediated Ca2+ flux may function as a way in which the plant can quickly sense external cues and promote internal signaling cascades, thereby regulating PCD.  We are only beginning to understand the interplay between plant immunity and meta-caspases following plant pathogen exposure, so investigating metacaspases in this context could help answer critical questions. For instance, could pathogen virulence factors suppress plant immune responses by specifically interfering with glutamate activity? Are metacaspases and their potential activation pathways specific targets for pathogens?”

Reviewer 3 Report

I have received a review Manuscript (ijms-2114381) entitled: ‘Regulating death and disease: exploring the roles of metacaspases in plants and fungi’ submitted to International Journal of Molecular Sciences.

In the submitted review the authors gathered and summarized recent advances in understanding of the divergence of metacaspase functionality in plants and fungi. The authors proved their excellent skill and knowledge in this scientific area, which is additionally supported by many scientific articles related to the elucidation of factors influencing death and disease in plants and fungi. I should be noted that scientific community considers this topic as highly valuable, there are other reviews in similar topic. Particularly, published few moths ago in IJMS article entitled “Evolutionary Diversity and Function of Metacaspases in Plants: Similar to but Not Caspases” (Int. J. Mol. Sci. 2022, 23(9), 4588; https://doi.org/10.3390/ijms23094588). In consequence, I would like to add this reference in the introduction and clearly indicate, the purpose of the submitted review in the context to available state-of-the-art (particularly other review articles published within last 10 years).

I noticed that all paragraphs are elaborated with care and do not bring any confusion to reader. I strongly agree with provided crucial conclusions, like “On the plant side, metacaspases can be critical in the prevention of pathogenicity, while on the fungal side, they can promote evasion of host defenses” or “Approaches that focus on the influence of metacaspases in disease incidence and prevention can provide a molecular basis of understanding virulence and defense mechanisms, ultimately increasing the production rate of globally relevant food sources.”

I also noted that authors remembered to include same basic information and terms, such aspathogen-associated molecular patterns (PAMPs) or effector-triggered immunity (ETI). In effect, future readers less familiar with this subject will be able to treat it as a very useful compendium allowing to summarize and deepen their knowledge. Below, I provide a list of elements that require correction/clarification before publication:

1.     Lines 18-21: in sentence “Bridging the gap between metacaspase roles in immunity and pathogenicity of plant pathogen interactions can enable more effective and targeted phytopathogen control efforts to increase production of globally important food crops, which is currently compromised by plant pathogens.” Part “which is currently compromised by plant pathogens.” is unnecessary.

2.     Lines 82-83: Sentence “As sessile organisms, plants must rely on cell signaling to perceive threats, alert neighboring cells of incoming attacks, and activate immune defenses” requires appropriate citation

3.     Lines 132-133: Sentence “Additionally, it is important to note that calcium is required for most plant metacaspase activity.” requires appropriate citation

4.     Lines 168-170: Sentence “By controlling the rate of Pep1 production, MC4 is able to provide a direct pathway for plant defense responses following both mechanical damage and pathogen invasion” has some editorial issues

Author Response

Response to Reviewer 3 Comments

I have received a review Manuscript (ijms-2114381) entitled: ‘Regulating death and disease: exploring the roles of metacaspases in plants and fungi’ submitted to International Journal of Molecular Sciences.

In the submitted review the authors gathered and summarized recent advances in understanding of the divergence of metacaspase functionality in plants and fungi. The authors proved their excellent skill and knowledge in this scientific area, which is additionally supported by many scientific articles related to the elucidation of factors influencing death and disease in plants and fungi. I should be noted that scientific community considers this topic as highly valuable, there are other reviews in similar topic. Particularly, published few months ago in IJMS article entitled “Evolutionary Diversity and Function of Metacaspases in Plants: Similar to but Not Caspases” (Int. J. Mol. Sci. 2022, 23(9), 4588; https://doi.org/10.3390/ijms23094588). In consequence, I would like to add this reference in the introduction and clearly indicate, the purpose of the submitted review in the context to available state-of-the-art (particularly other review articles published within last 10 years).

We thank Reviewer 3 for their efforts in reviewing our manuscript. We agree with this suggestion to incorporate this recent review into our manuscript. Please see lines 59-62 where this review article was added as reference #18.

I noticed that all paragraphs are elaborated with care and do not bring any confusion to reader. I strongly agree with provided crucial conclusions, like “On the plant side, metacaspases can be critical in the prevention of pathogenicity, while on the fungal side, they can promote evasion of host defenses” or “Approaches that focus on the influence of metacaspases in disease incidence and prevention can provide a molecular basis of understanding virulence and defense mechanisms, ultimately increasing the production rate of globally relevant food sources.”

I also noted that authors remembered to include same basic information and terms, such as pathogen-associated molecular patterns (PAMPs) or effector-triggered immunity (ETI). In effect, future readers less familiar with this subject will be able to treat it as a very useful compendium allowing to summarize and deepen their knowledge. Below, I provide a list of elements that require correction/clarification before publication:

  1. Lines 18-21: in sentence “Bridging the gap between metacaspase roles in immunity and pathogenicity of plant pathogen interactions can enable more effective and targeted phytopathogen control efforts to increase production of globally important food crops, which is currently compromised by plant pathogens.” Part “which is currently compromised by plant pathogens.” is unnecessary.

We thank Reviewer 3 for their editorial suggestion of the abstract. We agree with the suggestion to omit the latter portion of the sentence for clarity. Please find lines 18-21 show the updated change of the sentence “Bridging the gap between metacaspase roles in immunity and pathogenicity of plant pathogen interactions can enable more effective and targeted phytopathogen control efforts to increase production of globally important food crops, which is currently compromised by plant pathogens.” to “Bridging the gap between metacaspase roles in immunity and pathogenicity of plant pathogen interactions can enable more effective and targeted phytopathogen control efforts to increase production of globally important food crops.”.

  1. Lines 82-83: Sentence “As sessile organisms, plants must rely on cell signaling to perceive threats, alert neighboring cells of incoming attacks, and activate immune defenses” requires appropriate citation

We thank Reviewer 3 for pointing out our oversight in not including appropriate citations in this section. Please see line 88 where we have added the proper references, #21, 22.

  1. Lines 132-133: Sentence “Additionally, it is important to note that calcium is required for most plant metacaspase activity.” requires appropriate citation

We thank Reviewer 3 for pointing out our oversight in not including appropriate citations in this section. Please see line 138 where we have added the proper references, #37, 38.

  1. Lines 168-170: Sentence “By controlling the rate of Pep1 production, MC4 is able to provide a direct pathway for plant defense responses following both mechanical damage and pathogen invasion” has some editorial issues

We thank Reviewer 3 for the editorial suggestion on this sentence. Please see lines 183-185 that change the sentence “By controlling the rate of Pep1 production, AtMC4 is able to provide a direct pathway for plant defense responses following both mechanical damage and pathogen invasion.” to “AtMC4 is able to provide a direct pathway for plant defense responses following both mechanical damage and pathogen invasion through controlling the rate of Pep1 production.” for added clarity.